# Influence of Cluster Sets on Mechanical and Perceptual Variables in Adolescent Athletes

**DOI:** 10.3390/ijerph20042810

**Published:** 2023-02-05

**Authors:** Gustavo Api, Rosimeide Francisco dos Santos Legnani, Diogo Bertella Foschiera, Filipe Manuel Clemente, Elto Legnani

**Affiliations:** 1Department of Physical Education, Federal University of Technology, Curitiba 81310-900, Paraná, Brazil; 2Department of Physical Education, Ponta Grossa State University, Ponta Grossa 84030-900, Paraná, Brazil; 3Physical Education Collegiate, Federal Institute of Paraná, Palmas 85555-000, Paraná, Brazil; 4Escola Superior Desporto e Lazer, Instituto Politécnico de Viana do Castelo, Rua Escola Industrial e Comercial de Nun’Álvares, 4900-347 Viana do Castelo, Portugal; 5Instituto de Telecomunicações, Delegação da Covilhã, 1049-001 Lisboa, Portugal

**Keywords:** intra-set rest, resistance training, rest redistribution, young athletes, jump performance, back squat, rating of perceived exertion, Delayed Onset Muscle Soreness

## Abstract

Cluster sets (CS) are effective in maintaining performance and reducing perceived effort compared to traditional sets (TRD). However, little is known about these effects on adolescent athletes. The purpose of this study was to compare the effect of CS on the performance of mechanical and perceptual variables in young athletes. Eleven subjects [4 boys (age = 15.5 ± 0.8 years; body mass = 54.3 ± 7.0 kg; body height = 1.67 ± 0.04 m; Back Squat 1RM/body mass: 1.62 ± 0.19 kg; years from peak height velocity [PHV]: 0.94 ± 0.50) and 7 girls (age = 17.2 ± 1.4 years; body mass = 54.7 ± 6.3 kg; body height = 1.63 ± 0.08 m; Back Squat 1RM/body mass: 1.22 ± 0.16 kg; years from PHV: 3.33 ± 1.00)] participated in a randomized crossover design with one traditional (TRD: 3 × 8, no intra-set and 225 s interest rest) and two clusters (CS1: 3 × 2 × 4, one 30 s intra-set and 180 s inter-set rest; and CS2: 3 × 4 × 2, three 30 s intra-set and 90 s inter-set rest) protocols. The subjects were assessed for a Back Squat 1RM for the first meet, then performed the three protocols on three different days, with at least 48 h between them. During experimental sessions, a back squat exercise was performed, and mean propulsive velocity (MPV), power (MPP), and force (MPF) were collected to analyze performance between protocols, together with measures of countermovement jump (CMJ) and perceptual responses through Rating of Perceived Exertion for each set (RPE-Set) and the overall session (S-RPE), and Muscle Soreness (DOMS). The results showed that velocity and power decline (MVD and MPD) were favorable for CS2 (MVD: −5.61 ± 14.84%; MPD: −5.63 ± 14.91%) against TRD (MVD: −21.10 ± 11.88%; MPD: −20.98 ± 11.85%) (*p* < 0.01) and CS1 (MVD: −21.44 ± 12.13%; MPD: −21.50 ± 12.20%) (*p* < 0.05). For RPE-Set, the scores were smaller for CS2 (RPE8: 3.23 ± 0.61; RPE16: 4.32 ± 1.42; RPE24: 4.46 ± 1.51) compared to TRD (RPE8: 4.73 ± 1.33; RPE16: 5.46 ± 1.62; RPE24: 6.23 ± 1.97) (*p* = 0.008), as well as for Session RPE (CS2: 4.32 ± 1.59; TRD: 5.68 ± 1.75) (*p* = 0.015). There were no changes for jump height (CMJ: *p* = 0.985), and the difference between time points in CMJ (ΔCMJ: *p* = 0.213) and muscle soreness (DOMS: *p* = 0.437) were identified. Our findings suggest that using CS with a greater number of intra-set rests is more efficient even with the total rest interval equalized, presenting lower decreases in mechanical performance and lower perceptual effort responses.

## 1. Introduction

Resistance training is an integral part of athletic development for many individuals of various ages and sports [1]. It is well established in the literature that increasing strength and power capacity is an important strategy for improving sports performance [2] and injury prevention [3], and that higher levels of strength and power promote greater tolerance to training loads [4]. Nevertheless, acute effects of resistance training sessions could cause not only potential benefits, like post-potentiation activation, but can decrease performance as well. Several authors have discussed these effects on agility, endurance, power, jump height, and even in cognitive performance [5,6,7,8,9,10], which can be explained by the accumulation of fatigue due to higher exercise effort or even the proximity to failure, in which sets performed to failure have been shown to acutely impair jump and sprint performance compared to sets with lower velocity loss [11].

The use of different set configurations, with a reorganization of rest intervals, as cluster sets, appears to be a great strategy to help maintain the performance of variables such as force, power, and velocity with less development of fatigue indicators (e.g., perceived effort and jump performance) during the training session [12,13,14,15,16,17,18]. Cluster Sets (CS) are characterized by dividing sets into blocks of fewer repetitions with the addition of short intra-set intervals or the redistribution of intervals between repetitions [19,20,21,22]. The initial premise is that this configuration would provide a better increase in exercise quality with the ability to maintain or improve performance [13] and allow higher loads, leading to greater adaptations for performance [22].

Several studies have identified that, compared to the traditional configuration, cluster sets allow for greater maintenance of performance and less decline in power and velocity loss during the sets [12,13,14,16,17,23]. Tufano et al. [23] reported velocity and power declines to 23% on traditional sets and 1–5% in cluster sets configurations in back squat exercises for three sets of 12 repetitions in 60%1RM, demonstrating a better efficiency for cluster sets. Likewise, Cuevas-Aburto et al. [24] found lower mean velocity values for traditional sets compared to cluster sets, with about a 5% difference for both back squats and bench presses, using three sets of six repetitions against a 10RM load. In a study verifying neuromuscular fatigue through electromyography, it was evidenced that individuals who performed the Cluster configuration presented a reduced increase in electromyographic amplitude along with a lower reduction in frequency [25], pointing to a lower accumulation of fatigue. There was also less accumulation of metabolic substrates verified through the reduction in lactate, ammonia, and cortisol concentrations [17,26,27,28,29]. González-Hernández et al. [27] verified that besides the benefits of maintenance of performance in mechanical variables and less metabolic accumulation during exercise, there was also an association between more extended inter-repetition interval periods and lower perceptual responses measured through perceived effort, along with higher performance in vertical jump height. Regarding the rating of perceived exertion, lower scores were reported when there are greater frequencies of intra-set or inter-repetition intervals [18,30,31,32].

However, with more inter-repetitions or intra-set intervals, a longer session duration could also occur [33]. Torrejón et al. [34] found that when the work-to-rest ratio is equated with both the total volume of repetitions and the total rest intervals, the reorganization of rest intervals, specifically configuring short intra-series intervals, does not allow for higher movement velocities to be achieved compared to the traditional configuration. Still, Piqueras-Sanchiz et al. [10] observed that jump height performance is less attenuated (~10%) when greater inter-set rest interval frequencies are adopted during squat exercises when compared to traditional sets, even with equalized work-to-rest ratio and total training volume. Thus, the equalization of total volumes and intervals between protocols would be necessary for an effective comparison between configuration models, considering different formats of sets and rest interval structures. In addition, there is a research gap in using these configurations in younger and adolescent athletes, whose beneficial effects of resistance training using cluster sets are well-known in adult population samples consisting primarily of male subjects [14,17,35,36]. As little is known about its effects on young athletes [33,37,38], and as children and adolescents have a higher rate of recovery between sets of exercises compared to adults, consequent to lower production of power [39,40], it is justified to investigate the effects of cluster sets on this population, since the outcome concerning fatigue and maintenance of performance, in addition to the psychophysiological responses, may be divergent. In addition, there is an urge to address resistance training strategies that can reduce the risk of injury in this population, by allowing a smaller accumulation of fatigue and providing better adherence to training programs, since there are several benefits for health and performance listed in the literature, as well for psychosocial aspects such as self-confidence, body image, and socialization skills [41,42,43].

With this in mind, the main objective of this study was to compare the effect of a traditional set protocol (TRD: no intra-set rest) and two different cluster sets protocols (CS1: one intra-set rest; CS2: three intra-set rests) on mechanical, perceptual, and jump performance variables in adolescent athletes. As the main hypothesis, it is expected that CS configurations would be more efficient for minor decreases and better maintenance in mechanical variables, and attenuate the decrease in CMJ performance with lower scores for perceptual variables.

## 2. Materials and Methods

### 2.1. Study Population

The sample size was calculated a priori using the GPower v 3.1.9.6 (Heinrich-Heine-University, Dusseldorf, Germany) software [44], using the same parameters adopted by Ortega-Becerra, Sánchez-Moreno, and Pareja-Blanco [25], as a repeated measures ANOVA within factors, with Cohen’s effect size of 0.50 for comparison between protocols, error probability α = 0.05 and β = 0.95, resulting in a sample of 12 subjects. The sample was then selected by convenience, inviting subjects personally to one of the training practices at their respective sporting club. The subjects competing at state and national levels, free of injuries, from 13 to 19 years of age and with available time to participate were then selected for the study. However, there was a sample loss of one participant for not meeting the criteria of movement pattern execution during the squat, who, during the experimental sessions, could not reach parallel height for most of the repetitions during the execution of the squat and could introduce errors in the analysis; therefore, these data were then excluded from the study. Still, the sample resulted in a high sampling power (β = 0.93). Finally, 11 adolescent athletes consisting of 4 boys (age = 15.5 ± 0.8 years; body mass = 54.3 ± 7.0 kg; body height = 1.67 ± 4.06 m; 1RM load = 87.5 ± 13.2 kg; 1RM/body mass: 1.62 ± 0.19; years from peak height velocity [PHV]: 0.94 ± 0.50) and 7 girls (age = 17.2 ± 1.4 years; body mass = 54.7 ± 6.3 kg; body height = 1.63 ± 7.86 m; 1RM load = 66.6 ± 10.5 kg; 1RM/body mass: 1.22 ± 0.16; years from PHV: 3.33 ± 1.00), competing at the state or national level in their respective sports (badminton and volleyball), participated in the study. All participants had been practicing their respective sport three to five times a week, with two to three strength and conditioning sessions per week.

The criteria for participation consisted of at least six months prior experience with resistance training and being injury free for the last two months before the start of the study. The subjects needed to be able to perform a free-weight parallel back squat, in which the hip reached the height of the knees. Parents/guardians gave their written informed consent authorizing the minor-aged subject to participate in the study. This study was conducted under the approval of all ethical standards of the local University Ethics Committee and National Ethics Commission (protocol: 5.514.698), in line with the Declaration of Helsinki.

### 2.2. Study Design and Context

A randomized within-subjects cross-over study design was used to investigate the effects of three different set configurations on mechanical, perceptual, and jump performance variables. A repeated measures analysis of variance was conducted for mean propulsive velocity (MPV), power (MPP), and force (MPF) variables in the free-weight back squat exercise, for countermovement jump height (CMJ) before and after each protocol, for rating of perceived exertion, and Delayed Onset Muscle Soreness (DOMS). Performance decline and maintenance through the sets for mechanical variables were compared as well. The participants were assessed in four different sessions (Figure 1). The first session occurred one week before the first experimental intervention in the afternoon period (2:00 to 5:00 p.m.) and was used as a familiarization session, in which anthropometric measures were collected along with the assessment of physical variables [countermovement jump (CMJ) and the free-weight back squat one repetition maximum (1RM)], and presentation and explanation of the perceptual scales were provided with anchorage procedures by visual and verbal examples, and were used in the 1RM test for learning purposes. In the other three sessions, the participants were randomly allocated to one of the three different protocols of set configurations for three sets of eight repetitions at 75%1RM in the free-weight back squat exercise: a traditional set configuration protocol (TRD) consisting of two inter-set rest intervals of 225 s in between sets (3 × 8:225 s); a cluster set design (CS1) with one intra-set rest interval of 30 s within each set (fourth to the fifth repetition) and 180 s between sets (3 × 4:30 s:4:180 s); and another cluster set design (CS2) consisting of three intra-set rest intervals of 30 s within each set (second to third, fourth to fifth, and sixth to seventh repetitions) and 90 s between sets (3 × 2:30 s:2:30 s:2:30 s:2:90 s). Each protocol contained a total rest interval of 450 s, only differing by the rest configuration between and within sets. This rest redistribution was organized so that total rest intervals were equalized between the three experimental protocols.

The participants performed each protocol in a random order, with recovery criteria of at least 48 h between experimental sessions (mean ± standard deviation: 5.2 ± 3.4 days), a minimum of 15 points (well recovered) on the total quality of recovery scale (TQR), and a muscle soreness maximum score of 3 points in a numeric rating scale for pain (NRS). If participants had not properly recovered, the experimental session would be postponed until the recovery criteria were met, allowing the participants to be in good condition and recovered for each day. For all protocols, the participants first reported their perceived recovery and made a general warm-up with mobility exercises, a set of 10 bodyweight squats, and a set of five submaximal CMJ; then, CMJ height was assessed before and after the back squat exercise. For the back squat, 50% and 70%1RM warmup sets for six and three repetitions were performed before selecting 75%1RM for the experimental protocol. Since fluctuations in 1RM occur day-to-day, the intensity was adjusted by the best mean propulsive velocity reached at 70%1RM using individual polynomial equations with the velocities and weights used during the back squat 1RM assessment, similar to those proposed by Thompson et al. [45], which were then calculated to determine the 75%1RM of the day. The participants then executed one of the mentioned protocols, reporting a rating of perceived exertion at the end of each set of eight total repetitions (RPE8, RPE16, and RPE24) and CMJ height was measured again right after the end of the third set, five and ten minutes later. Fifteen minutes after finishing the protocol, the participants reported the overall Session RPE (S-RPE). For indirect markers of muscle soreness, muscle pain was assessed using a bodyweight squat, asking for the intensity of pain between 1 and 10 points with an NRS, similar to the proposed by Doma et al. [5] at 24 and 48 h after the protocol. Figure 2 presents the stages of the experimental protocols.

The study took place during a short preparatory mesocycle during the months of July and August for 38 days, in which the subjects were participating in their daily sports training and school activities. During this period, the subjects performed the experimental sessions on separate days according to their availability of time, occurring during morning and afternoon periods (10:00 a.m. to 4:00 p.m.) but respecting the adopted criteria of at least a 48 h interval between protocols, in addition to the criteria of muscle pain and recovery. Although different days between subjects occurred, short adaptations from strength were controlled by a load-velocity profile, and the weight could be adjusted according to the mean propulsive velocity corresponding to the relative intensity, which has been reported as a tool for adjusting intensity by other studies [46,47,48]. As the study did not take place under laboratory conditions and due to the unavailability of athletes absent from training, the study was carried out at the training site during their available time for the scheduled strength conditioning sessions. However, it was not possible to control environmental conditions such as temperature and humidity, because the room did not have a system to condition the environment. In addition, the subjects had prior experience performing the free-weight back squat exercise in strength and conditioning sessions.

### 2.3. Anthropometrics and Peak Height Velocity

For the anthropometric measures, standing and sitting heights were measured through a wall stadiometer (Sanny^®^ ES2030, São Paulo, Brazil). For the standing height, subjects were barefoot and the evaluator positioned them right in the middle of the stadiometer with feet, hips, shoulders, and head touching the wall. For sitting height, the subjects were measured while seated in a 50 cm height box, with thighs parallel to the floor, and hips, shoulders, and head touching the wall; then, the height of the box was subtracted from the total seated measure. Body mass was assessed via an electronic scale (Filizola^®^, São Paulo, Brazil) with subjects centered in the middle of the scale, barefoot and wearing light clothes.

The maturity offset prediction equation provided by Mirwald et al. [49] was used to identify the age at peak height velocity (PHV), taking into account the chronological age (CA) body mass, standing and sitting height, and estimated leg length through standing height minus sitting height. The following equations were used for boys:Maturity offset = −9.236 + [0.0002708 × (Leg Length × Sitting Height)] + [−0.001663 × (CA × Leg Length)] + [0.007216 × (CA × Sitting Height)] + [0.02292 × (body mass by standing height ratio × 100)];(1)
and girls:Maturity offset: −9.376 + [0.0001882 × (Leg Length × Sitting Height)] + [0.0022 × (CA × Leg Length)] + [0.005841 × (CA × Sitting Height)] − [0.002658 × (CA × Body Mass)] + [0.07693 × (body mass by standing height ratio × 100)].(2)

### 2.4. Mechanical Variables

Measures of mean propulsive force (MPF), power (MPP), and velocity (MPV) were collected in the execution of all repetitions during the 1RM Back Squat Test and in the experimental sessions by a valid and reliable linear position transducer (LPT) (Chronojump^®^—Bosco System, Barcelona, Spain) [50,51], with a sampling frequency of 1000 Hertz and resolution of 1 mm. Through the LPT, velocity, and acceleration data were calculated from the displacement of the bar relative to time, the force was calculated as the mass lifted multiplied by the total acceleration (gravity + bar), and the power was calculated by the multiplication of force by velocity. In addition, the concentric phase of the motion in which the measured acceleration was greater than the acceleration due to gravity corresponded to the propulsive phase [52]. The sets and repetitions were compared between the variables in the protocols. Moreover, the effects of each configuration across every set during the protocols were assessed by a percentual decline of velocity (MVD), power (MPD), and force (MFD):Percent decline = [(repetition_last_ − repetition_first_)/repetition_first_] × 100(3)

In addition, maintenance percentage during the set of all mechanical variables (Velocity: MVM; Power: MPM; Force: MFM) was calculated as well, with the following equation, as proposed by Tufano et al. [23]:Maintenance_set_ = 100 − [(mean_set_ − repetition_first_)/repetition_first_ × 100(4)

Intraclass correlation coefficient presented excellent reliability with satisfactory coefficients of variation for MPV [ICC (95% CI): 0.983 (0.963–0.995); CV: 14%], MPP [ICC (95% CI): 0.995 (0.988–0.998); CV: 14%] and MPF [ICC (95% CI): 1.000 (1.000–1.000); CV: 3%] assessed through all repetitions between protocols.

### 2.5. Countermovement Jump

The vertical jump height was measured via flight time through CMJ using a contact platform (Jump System PRO—Cefise^®^—São Paulo, Brazil) with a sampling frequency of 1000 Hertz and a mean error of ±3 milliseconds. The participants were asked to stand upright with their hands at the waist and then perform a half squat to approximately 90 degrees of knee flexion, and without pause, perform the vertical jump. The participants were not allowed to move their arms or raise their knees towards the chest during the jump.

As a warm-up protocol, joint mobility exercises focused on the lower limbs and lumbar and thoracic spine were performed, followed by a set of ten bodyweight squats and five submaximal jumps. Subsequently, after a two-minute break, the participant performed five maximal jumps, with a ten-second interval between jumps. The average of the five jumps was used for the analysis, due to the average value being more sensitive to identify fatigue-induced changes [53]. This procedure was adopted in all experimental sessions before (CMJPre), immediately after (CMJ0′), five minutes after (CMJ5′), and ten minutes (CMJ10′) after performing the experimental protocol.

For the comparison between the pre- and post-moments of the vertical jump evaluation in all protocols, the calculation of the difference between the moments (ΔCMJ) was performed, returning its relative value in percentage, through the equation below:ΔCMJ = [(post − pre)/pre] × 100(5)

Reliability in the CMJ was realized through all pre-protocol jump measures for every experimental session, resulting in excellent reliability with a low coefficient of variation [ICC (95% CI): 0.996 (0.991–0.999); CV: 5%].

### 2.6. 1RM Back Squat Test

A progressive test for back squats was standardized for each individual in all sessions with the participants from a standing position with knees and hips fully extended, feet parallel with slight external rotation and approximately at shoulder width, and the bar supported on the back at the level of the acromion. Each participant performed a continuous downward movement up to the maximum possible amplitude in which there is no pelvic retroversion, so as not to compromise the quality of the exercise, but always reaching at least the parallel squatting position, and then returning to the initial position. The evaluation of the range of motion was performed by a qualified powerlifting coach, guaranteeing the established criteria. The eccentric phase velocity was controlled, adopting an average velocity of at least 0.4 m/s, with verbal feedback given to the participants when they did not reach the stipulated minimum velocity. At the end of the eccentric phase, there was a rapid transition to the concentric phase, which should be performed at the highest possible intentional velocity. The use of belts, knee pads, or any performance-enhancing device during the execution of the movement was not allowed.

The protocol adopted for the test was similar to that performed by Thompson et al. [45], consisting of a general joint warm-up and then the performance of the following relative intensities: five repetitions for 0% (body weight and a wooden stick), three repetitions for 30% and 50%, two repetitions for 70% and 80%, and one repetition for 90%, 95%, and 100%1RM, the latter with three attempts and increments from 2.0 to 5.0 kg. The test was terminated when the participant and the researcher agreed that a new attempt with more weight was not possible, or when two consecutive failures occurred between attempts to set the 1RM. An interval of three to five minutes was established between the attempts, and the predicted 1RM value was previously defined between the trainer and participant.

### 2.7. Velocity-Regulated Intensity

Since strength levels have daily fluctuations influenced by fatigue and readiness state [54,55], the relative intensity was adjusted by repetition velocity for every protocol. For this, data from the 1RM Back Squat test were used for a 2nd-order polynomial regression equation to determine the predicted 1RM, adopting the repetitions with the highest MPV of each relative intensity and the MPV in the 100%1RM trials as the minimum velocity threshold. The function “LINEST” with adjustment to the 2nd order was used in Microsoft Excel (Microsoft^®^, Redmond, WA, USA) to determine the parameters for calculating the equation:ax^2^ + bx + c = y(6)
where a, b, and c are the coefficients of the equation, and x was replaced by the mean propulsive velocity of the best repetition at 70%1RM (vel70%), resulting in the adjusted value of y, referring to the load equivalent to the execution velocity. Load intensity was readjusted to determine 75%1RM from the equivalent relative intensity calculated from the velocity.

### 2.8. Perceptual Variables

The following criteria developed for conducting the experimental sessions were previously collected for each protocol: the total quality recovery perceived scale (TQR) proposed by Kentä and Hassmén [56], and Delayed Onset Muscle Soreness (DOMS) through NRS similar to the scale proposed by Doma et al. [5] in the bodyweight squat. In TQR, the participants rated their psychophysiological recovery from the previous 24 h including the previous night’s sleep, on a scale of 6 (not recovered) to 20 (very well recovered). DOMS was assessed using a 1–10 numeric rating scale, with 1 defined as “no pain” and 10 as “very, very sore” and was also assessed 24 (DOMS24) and 48 (DOMS48) hours after the experimental sessions. The participants rated perceived exertion (RPE) through CR-10 [57], where 0 was defined as “rest” and 10 as “maximum effort”, for each set (RPE8, RPE16, RPE24) and for the session (S-RPE). The answers were conceded individually and recorded by the researcher in a purpose-made Microsoft^®^ Excel spreadsheet specially elaborated for the research.

### 2.9. Statistical Analysis

The data of the mechanical variables were stored in the Chronojump^®^ software database version 2.2.0 (Chronojump^®^ Bosco-system, Barcelona, Spain), to be later exported to a Microsoft Excel^®^ (Microsoft^®^, Redmond, WA, USA) spreadsheet and then grouped with the other variables for their subsequent tabulation and organization. Statistical analyses were performed using SPSS Statistics^®^ v.20.0 (IBM^®^, Armonk, NY, USA) software.

Descriptive data were reported using the means and standard deviations (SD). Data distribution was explored using the Shapiro–Wilk test. The reliability of the variables in each protocol was calculated by a two-way mixed effects model and average measures in an intraclass correlation coefficient (ICC) with a 95% confidence interval (95% CI), and the values were interpreted according to the guidelines presented by Koo and Li [58], in which values less than 0.5, between 0.5 and 0.75, between 0.75 and 0.9, and greater than 0.90 are indicative of poor, moderate, good, and excellent reliability, respectively. Coefficients of variation (CV) of values equal to or less than 15% were classified as satisfactory according to Stokes [59].

For comparison of the results of the means of each protocol, repeated measures analysis of variance (RM-ANOVA) was performed. For comparison of mechanical variables in the series performed between protocols (3 × 3, Protocol × Set), as well as for comparison between repetitions (3 × 24, protocol × repetitions), and comparison for RPE-Set (3 × 3, Protocol × Set), CMJ (3 × 4, Protocol × Time), ∆CMJ (3 × 3, Protocol × Time), and DOMS (3 × 2, Protocol × Time), a Two-Way RM-ANOVA was performed. When the assumption of sphericity was violated, the Greenhouse–Geisser correction was applied. Statistical adjustments by Bonferroni post hoc tests for the detection of differences between factors were performed for each of these analyses.

A probability value of *p* ≤ 0.05 was established. Effect sizes were measured using the Hegdes’ g [60], with the interpretation of the values proposed by Hopkins et al. [61] being: less than 0.2 = trivial; greater than 0.2 and less than 0.6 = small; greater than 0.6 and less than 1.2 = moderate; greater than 1.2 and less than 2.0 = large; greater than 2.0 and less than 4.0, very large; and greater than 4.0 = nearly perfect. In addition, RM-ANOVA effect sizes through partial eta squared (η_p_^2^) were reported.

## 3. Results

### 3.1. Criteria

For the recovery criteria, there was no difference between protocols for TQR [F(1.34:13.40) = 0.365; *p* = 0.618; CS2: 16.91 ± 1.88; CS1: 17.00 ± 1.60; TRD: 17.45 ± 1.78] and DOMS [F(2:20) = 0.48; *p* = 0.953; CS2:1.82 ± 0.72; CS1: 1.83 ± 0.74; TRD 1.91 ± 0.67]. However, for the session time duration, as rest intervals were equalized between conditions since cluster configurations may take more total duration time if they were not equalized, there was a significant difference between protocols [F(2:20) = 4.096; *p* = 0.032]. However, post hoc tests showed that CS2 (8.45 ± 0.13 min) had a significantly shorter duration compared to TRD (8.52 ± 0.11 min).

### 3.2. Velocity, Power, and Force Variables

Two-Way RM-ANOVA (Protocol × Set) found significant differences in the main effect between protocols for the variables of MPV [F(2:20) = 9.04, *p* = 0.002; η_p_^2^: 0.475], MPP [F(2:20) = 10.205, *p* < 0.01, η_p_^2^: 0.505], MVD [F(2:20) = 12.745, *p* < 0.01, η^2^: 0.560], MPD [F(2:20) = 12.567, *p* < 0.01, η_p_^2^: 0.557], MVM [F(2:20) = 13.9, *p* < 0.01, η_p_^2^: 0.582] and MPM [F(2:20) = 13.651, *p* < 0.01, η_p_^2^: 0.577], with Bonferroni’s post-hoc test showing significant differences for all of these variables, in favor of the CS2 protocol compared to TRD (MPV: *p* = 0.007; MPP: *p* = 0.006; MVD: *p* = 0.006; MPD: *p* = 0.006; MVM: *p* = 0.003; MPM: *p* = 0.003).

Post-hoc analyses also identified a statistically significant difference between protocols CS2 × CS1, for the variables MVD, MPD (both *p* = 0.007), MVM, and MPM (both *p* = 0.014), but identifying no differences between CS1 × TRD. There were no statistical differences across protocols for MPF, MFD, and MFM. The mean ± SD and effect sizes for the mechanical variables are presented in Table 1.

Figure 3 reports the individual measures comparisons between protocols for mean propulsive velocity (left) and mean propulsive power (right), using the traditional protocol (TRD) as the reference model for comparison with the Cluster 1 (CS1) and Cluster 2 (CS2) protocols.

There was also a main effect between sets for the variables MPV [F(1.26:12.63) = 11.232, *p* = 0.004, η_p_^2^: 0.529] and MPP [F(1.21: 12.08) = 10.437, *p* = 0.005, η_p_^2^: 0.511], pointing to statistically significant post hoc differences between 1st and 2nd (*p* = 0.032), and 1st and 3rd (*p* = 0.015) sets for MPV, and 1st and 2nd (*p* = 0.042), 1st and 3rd (*p* = 0.02), and 2nd and 3rd (*p* = 0.05) sets for MPP. No main effect was identified for any of the other mechanical variables. In addition, no Protocol × Set interaction effect was identified for any of the mechanical variables.

For the comparison between protocols and repetitions, the Two Way RM-ANOVA (Protocol × Repetitions) (10 participants), showed a significant main effect of Protocol on MPV [F(2: 18) = 10.113, *p* = 0.001, η_p_^2^: 0.529] and MPP [F(2:18) = 11.093, *p* = 0.001, η_p_^2^: 0.552] and post-hoc differences between protocols CS2 × TRD (MPV: *p* = 0.009; MPP: *p* = 0.008) and CS1 × TRD (MPV: *p* = 0.007; MPP: *p* = 0.011). There was also a main effect for Repetitions on MPV (F(4.60:41.37) = 10.817, *p* < 0.001, η_p_^2^: 0.546] and MPP [F(3.88:34.93) = 10.999, *p* < 0.001, η_p_^2^: 0.550] with post-hoc differences for MPV between: 3rd × 8th, 14th, 15th, 16th, and 24th (*p* < 0.05); 10th × 16th (*p* = 0.018); and 5th, 9th, 11th and 17th × 24th (*p* < 0.05). In addition, there were post-hoc differences for MPP between: 3rd × 8th (*p* = 0.045); 11th × 8th, and 24th (*p* < 0.05). No significant differences were found for MPF. In addition, no Protocol × Repetitions interaction was found for MPV, MPP and MPF. Figure 4 and Figure 5 shows the means and standard error for repetitions between protocols for MPV and MPP, respectively.

### 3.3. Jump Variables

No statistically significant differences were identified between protocols for vertical jump height [F(2:20) = 0.015, *p* = 0.985; η_p_^2^: 0.002] nor ΔCMJ [F(2:20) = 1.671, *p* = 0.213, η_p_^2^: 0.143]. However, there were differences between times for CMJ height [F(3:30) = 18.346, *p* < 0.001, η_p_^2^: 0.647], with post-hoc identifying differences between times CMJPre × CMJ0’ (*p* = 0.02), CMJPre × CMJ5’ (*p* < 0.001), and CMJPre × CMJ10’ (*p* < 0.001). No difference between times was identified for ΔCMJ [F(1.20:12.00) = 3.588; *p* = 0.077, η_p_^2^: 0.264]. No Protocol × Time interaction occurred for any of the variables related to the vertical jump. The jump performance data are presented in Table 2.

### 3.4. Perceptual Variables

For the RPE-Set in the protocols, a statistically significant main effect was identified for Protocol [F(2:20) = 7.721, *p* = 0.003, η_p_^2^: 0.436] and Set [F(1.23:12.30) = 18.412, *p* = 0.001, η_p_^2^: 0.648]. For the Protocol factor, the post hoc analysis only showed a difference between CS2 and TRD (*p* = 0.008), and for the Set factor, there was a difference between all series, where the 1st set was lower than the 2nd set (*p* = 0.01) and 3rd set (*p* = 0.003), and the 2nd set was lower than the 3rd set (*p* = 0.008). There was no Protocol × Set interaction for the RPE-Set across protocols. Similarly, for session RPE (S-RPE), a significant difference was observed between protocols [F(2:20) = 7.694, *p* = 0.003, η_p_^2^: 0.435], with the post-hoc favoring the CS2 protocol over the TRD (*p* = 0.015). Figure 6 presents individual measures of the RPE-Set scores for the sets of each protocol. Table 3 presents the data for RPE-Set and Session-RPE between protocols and time points with their respective effect sizes.

The Delayed Onset Muscle Soreness scores were lower for CS2 (DOMS24: 3.59 ± 1.86; DOMS48: 2.82 ± 1.60) and CS1 (DOMS24: 3.77 ± 2.07; DOMS48: 2.86 ± 1.95), compared to TRD (DOMS24: 4.32 ± 2.00; DOMS48: 3.68 ± 2.13); however, no significant differences were found between protocols [F(2:20) = 0.863, *p* = 0.437, η_p_^2^: 0.079], showing small effects sizes for comparisons between CS2 × TRD (DOMS24: 0.36; DOMS48: 0.44) and CS1 × TRD (DOMS24: 0.26; DOMS48: 0.39), and negligible effects for CS2 × CS1 (DOMS24: 0.09; DOMS48: 0.02). There was a significant difference between time points [F(1:10) = 9.996, *p* = 0.01, η_p_^2^: 0.500] where DOMS24 showed higher pain scores compared to DOMS48 (*p* = 0.01). There was no Protocol × Time interaction.

## 4. Discussion

The main hypothesis of this study was that cluster set configurations could be effective in maintaining performance in mechanical, jumping, and perceptual variables. The hypothesis was partially confirmed since not all cluster configurations caused the same expected effect. However, in conformity to other studies, a higher frequency of intra-set rest intervals was shown to be more efficient in maintaining mechanical performance and for lower ratings of perceived effort during exercise.

Our results showed that mean propulsive velocity (MPV) and mean propulsive power (MPP) were different only for CS2 in comparison to TRD, showing no differences between CS1 with TRD. As well, mean propulsive velocity and power presented smaller decreases and better maintenance, especially when more intra-set rest intervals were applied on the sets, showing that CS2 was superior to CS1 and TRD. However, CS1 was not different from TRD. These results are in concordance with other studies, which showed that more frequent interest rest intervals could produce better performance for these variables [18,23,24,62,63]. Similar to Tufano et al. [18,23,62], Wetmore et al. [36], and Oliver et al. [64], the results regarding mean or average concentric velocity in back squats favored CS configurations over TRD, allowing the maintenance of better performance throughout the exercise. However, in contrast to these works, our results did not show the superiority of the one intra-set rest interval (CS1) protocol compared to the traditional configuration (TRD) for the variables above. One possible reason is the total rest intervals were equalized, which could have affected the results, in agreement with similar study designs of other researchers [34,65]. According to this, an increase in the number of rest periods may enhance recovery through the maintenance of phosphocreatine (PCr) and adenosine triphosphate (ATP) stores and increased metabolite clearance (e.g., lactate accumulation) [66], which could permit a higher substrate availability, allowing the maintenance of movement velocity across all sets [67]. However, there was only 45 s difference in inter-set rest interval between TRD and CS1 (TRD: 225 s; CS1: 180 s), possibly causing these changes to be less pronounced since there was sufficient time in TRD to allow a similar recovery. It can be hypothesized that because the majority of subjects in this sample were mature adolescents girls and recent research has shown that women could be less influenced by short rest intervals in comparison to men [68], there were smaller decreases in movement velocity in back squats (although, men had higher movement velocity) and also lower accumulation of blood lactate, which is related to similar ratings of velocity loss [17]. Therefore, CS1 and TRD had identical values for MPV and MPP declines, possibly explained by this factor.

The force variables did not show differences between the protocols. Although the load was adjusted by velocity, there were no statistical differences between conditions and between sets, so changes in MPF did not occur since the loads were statistically equal. Latella et al. [38] reported in a meta-analysis that, overall, studies show that CS does not have an effect on MPF. However, it had been reported that CS might reduce losses in peak force [23,64], explaining that the changes in MPP were mainly influenced by movement velocity.

Unexpectedly, jump performance did not show statistically significant changes between protocols either for CMJ or ΔCMJ, as hypothesized. However, for CS2 and CS1 × TRD, moderate effect sizes were demonstrated between conditions (CS2-TRD: 0.81; CS1-TRD: 0.84) when comparing ΔCMJ for Pre-0′. These finds were contrary to the results from Girman et al. [26] and Varela-Olalla et al. [29]. However, different from our study, Girman et al. [26] used two exercises and two circuits in their design, which could have enhanced fatigue for the traditional configuration. On the other hand, the study by Varela-Olalla et al. [29] was not randomized since the design required participants to reach 20% velocity loss in the traditional configuration session, then they accounted for the same number of repetitions for the cluster configuration, in addition to the use of half squat exercises. Moreover, different relative intensities were used. However, similar results were reported by Cuevas-Aburto et al. [24], in which CS and TRD configurations induced comparable decreases in jump performance (from −6.0% to −8.1% vs. from −5.3% to −8.0% in post ten minutes measures for both studies). Despite CS having a higher training duration, since inter-set rest was the same between the configurations (except for RR), and the study design accounting for bench press exercises in the same session, these results are aligned with the present study, as similar relative intensities were used (10RM, equivalent to 75%1RM [69]), in contrast to the studies above where smaller relative intensities were used (~60–65%). According to the authors [18,24,26], the lack of differences in CMJ could be due to the fast drop of metabolic fatigue after the protocol, independently of the type of configuration. However, as shown in the present study, the ΔCMJ for Pre-0′ was higher for CS2 and CS1 compared to TRD than in Pre-5′ and Pre-10′. Although not significant, Protocol × Time interaction for CMJ and ΔCMJ were almost significant (*p* = 0.052 and *p* = 0.057, respectively), speculating that the results might have been different if the sample size was larger. Despite this, the adolescents present higher recovery capacity between sets in resistance training sessions compared to adults [40], which may have produced a faster recovery for the TRD configuration since there was sufficient time for recovery and, therefore, did not show statistical differences when compared to CS.

Lower scores of RPE in CS2 against TRD were found in the present study between sets and for RPE-S, but no difference was identified for CS1 with CS2 or TRD. These finds agree with other studies, which reported comparing CS and TRD configurations [27,30,70]. In comparison to Cuevas-Aburto et al. [24], the results regarding RPE in sets between protocols were similar, which found higher values after the sets for TR (SQ: 6.9 ± 0.7) compared to CL (SQ: 6.2 ± 0.8) and RR (SQ: 6.2 ± 0.8). However, the average RPE-Set scores for the present study were smaller (CS2: 4.00 ± 1.33; CS1: 5.08 ± 2.24; TRD: 5.47 ± 1.72) but with higher SD. While the session RPE in the study, as mentioned above, was not significantly different between the set configurations (*p* = 0.595), we found higher S-RPE values for TRD compared to CS2 but not CS1. Although the RPE values for adolescents and adults could be different, this could be explained by most participants not reaching failure during exercise; in fact, only one participant reached failure during one set in the study. Another reason could be related to the ability of young athletes to self-assess their perception of load and effort, which could be unreliable [71]; nevertheless, to control for this bias, anchoring procedures were used during the 1RM Back Squat Test, and the athletes had previous experience reporting RPE in their daily sports training. Since the strength adaptations between the TRD and CS configurations are similar [33], these results suggest that CS can be used as a strategy to induce less psychophysiological fatigue.

Unexpected, DOMS were not different between the conditions. Although the order of protocols was randomized, no significant changes were observed. These findings are in agreement with the results of Varela-Olalla et al. [29], even though higher scores were reported for TRD and without sample randomization, no difference between conditions was presented. However, Merrigan et al. [31], comparing rest-redistribution and traditional protocols across several times points (pre, post, 24, 48, 72, and 96 h) in a randomized, counterbalanced, repeated measures design also did not find statistical differences for muscle soreness between conditions at any time points. In the current study, when analyzing the applied protocols order instead of the structure of set configuration, a statistical difference (*p* < 0.01) was found between the time points, showing higher scores for the first protocol compared to the second and the third (*p* < 0.01) and the second protocol was higher than the third one (*p* = 0.01). While not considered for the current study analysis, this information helps to understand limitations, such as the influence of the repeated bout effect [72]. Long interspersed sessions of resistance training, varying from 10 days [72] to 4 weeks [73,74], have been shown to diminish the effects of indirect markers of muscle damage, in which scores of muscle soreness are lower for the last bout compared to the first ones, implying that adaptation to the exercise had occurred during the subsequent bouts. Caution must be taken when extrapolating the results of the current study since these did not use CS configurations or similar exercise conditions. However, even with the load adjusted for every protocol to control for these adaptations, another external factor may have contributed, such as training load derived from sport-specific training, sleep, or psychological stress.

Some limitations could be drawn from the current research. The present study was not performed in laboratory conditions with the back squat exercise performed in a Smith Squat Machine, which could prevent changes in movement pattern derived from horizontal displacement, and thus unfortunately losing internal validity; however, the study was performed with athletes’ practicing their current daily activities on their respective sports modalities and using free-weight back squats in strength and conditioning sessions, which could have influenced the results since they are in their real conditions, enhancing external validity and being able to extrapolate for real-world training settings. Although recovery criteria were adopted to prevent the influence of external factors beyond the protocols, the rest days varied between the subjects due to muscle pain impairment that was possibly accumulated from their specific sports training, which was not controlled or monitored. In addition, some athletes had scheduled friendly matches which caused the need to postpone some experimental sessions. Exercise volume and intensity on the court, as well nutrition status, were not controlled, which could have impaired recovery between days. In addition, three subjects performed the protocols in the morning period, while the remaining participants performed in the afternoon, so the exercise and jump performance may have been influenced by circadian rhythms. It would be advisable to monitor sports practice training loads, and control for covariates, as well as to establish a more reliable margin of rest days; despite this, these limitations reflect the challenges of researching real-world settings in a sports training context. Therefore, more robust designs may be implemented with repeated measures for the same conditions or between-subjects designs to increase comprehension of these results.

Regarding the sample size, although it was calculated a priori, resulting in a high power (0.93) even with only 11 subjects, sample or data loss is still a problem. As for MPV and MPP, the results from the interaction analysis for CMJ and ΔCMJ should be different if the sample size were bigger. CMJ and ΔCMJ presented almost significant differences for Protocol × Time interaction, as well as a tendency for a significant main effect in Time for ΔCMJ. Since one subject’s data were excluded from repetitions due to failure before completing the last set in the CS1 protocol, the analysis for Protocol × Repetitions interactions for MPV and MPP would be different. The authors did an imputation method for the missing value to verify if it would occur differently from the actual analysis. The criteria for imputation must be respected, although there is no reference cut-off value regarding the margin of acceptance of missing values in the literature [75], the rule of thumb is that when the rate of missing information is below 5%, single-imputation inferences may be fairly accurate [76]. However, in contrast, Bennett [77] affirms that statistical analysis may be biased if missing data are greater than 10%. However, the current missing value analyses were 9.1% for between-subjects data (1 out of 11 subjects) and 0.1% for within-subjects data (1 out of a total of 72 repetitions). Considering this information and under these conditions, a simulation for a more optimistic scenario was used to compare the results. If a missing value was imputed (single imputation from the subjects mean), the Protocol × Repetitions interaction would be statistically significant for MPV [F(7.47:74.67) = 2.254, *p* = 0.036, η_p_^2^ = 0.184, Observed Power: 0.819] and MPP [F(7.14:71.42) = 2.202, *p* = 0.043, η_p_^2^ = 0.180, Observed Power: 0.794), but no difference for MPF [F(2.58:25.79) = 0.922, *p* = 0.432, η_p_^2^ = 0.084, Observed Power: 0.211) would be observed.

This research accounted for comparing these variables between matured boys and girls, which could confound the results since there is evidence for differences between the sexes [68]. However, regarding the level of strength and training experience, although the athletes had a large variation in relative strength (range: 1.0 to 1.8 BW/Kg in back squats) and all had more than 6 months of resistance training experience, there is evidence that there are no statistical differences influenced by these factors [33,38,66].

Nonetheless, when considering the literature reviewed by the authors of this study, the present study was the first to compare the acute effects of different CS and TRD protocols for the free-weight back squat exercise on adolescent athletes. Several strengths must be highlighted in the research. When equalizing total rest intervals, no major changes occurred between CS1 and TRD, as more frequent intra-set rest intervals (CS2) were needed to decrease the losses in velocity and power [64,78], as well as for smaller RPE scores [65]. Although no statistical difference for CMJ performance were observed, CS resulted in lower decrements on ΔCMJ when comparing pre- to immediate post-measures, as shown by the moderate effect sizes. Therefore, CS configurations seem to be effective in acutely reducing mechanical fatigue during back squats for velocity and power, allowing better maintenance and reducing perceived effort for the entire set. When taking into account specific sport performances (e.g., jump height), CS may not be a superior strategy to TRD; however, more research is needed to elucidate this question. Likewise, no difference has been identified for DOMS, and although not significant, CS2 showed smaller mean scores compared to TRD. Future studies should investigate CS configurations concerning different lengths of intra-set and inter-set rest intervals, training volumes, load intensities, and exercises for adolescents in a wide range of sports and verify the influence of fatigue for sport-specific variables, controlling for external factors such as training load and carrying out longitudinal analyses for strength, power, and endurance adaptations.

Practical applications for training periodization could be drawn from these inferences. The use of CS seems to be a good strategy for the maintenance of technical proficiency, allowing a better movement quality with lower fatigue [15]. These set configurations could be important for a preparatory period in which the aim is to enhance work capacity without decreases in technical performance. The same is true for the special preparation period where power development is the main objective, and where CS could allow better power performance compared to more traditional methods, even when long rests are prescribed [79]. These findings could also be applied in competition periods when low levels of fatigue are desired. Overall, CS may be effective in helping to diminish internal training loads considering the total load added from specific sports training (e.g., TRIMP, Session-RPE).

## 5. Conclusions

As presented by the current study, CS with more frequent intra-set rest intervals seems to be an effective strategy to promote better maintenance of mechanical variables across all sets for back squat exercises in adolescent athletes, as well as inducing lower ratings of perceived effort compared to TRD. However, no differences were shown in jump performance and muscle soreness. Strength and Conditioning coaches could apply these configurations to manage fatigue across the training session, decreasing internal loads and increasing velocity and power performance in back squats without compromising total training time.

## Figures and Tables

**Figure 1 ijerph-20-02810-f001:**
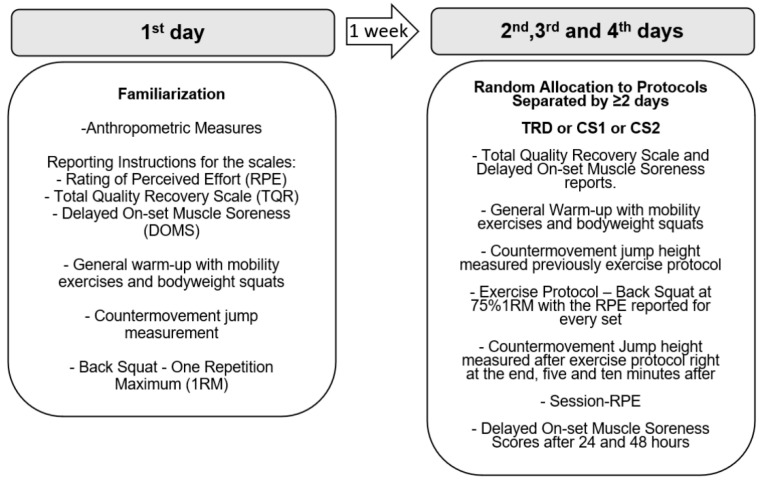
Summary of the study. 1st day refers to the familiarization session while 2nd, 3rd, and 4th days refers to the experimental sessions. Traditional (TRD), Cluster 1 (CS1), and Cluster 2 (CS2) set configurations for the back squat exercise.

**Figure 2 ijerph-20-02810-f002:**
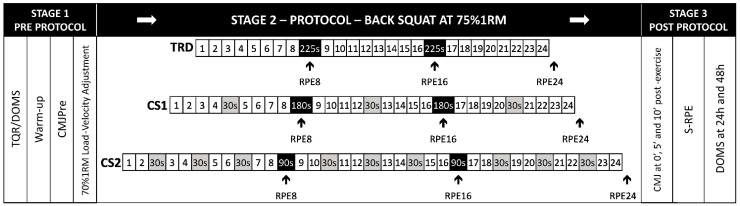
Study design—Traditional (TRD), Cluster 1 (CS1), and Cluster 2 (CS2). TQR: Total Quality Recovery Scale; DOMS: Delayed Onset Muscle Soreness (pre-protocol, 24 and 48 h post protocol); CMJ: jump height before (CMJPre) and after (CMJ at 0′, 5′, and 10′) protocol. RPE8, 16, 24: Rating of Perceived Exertion at the end of each set; S-RPE: Rating of Perceived Exertion for overall session.

**Figure 3 ijerph-20-02810-f003:**
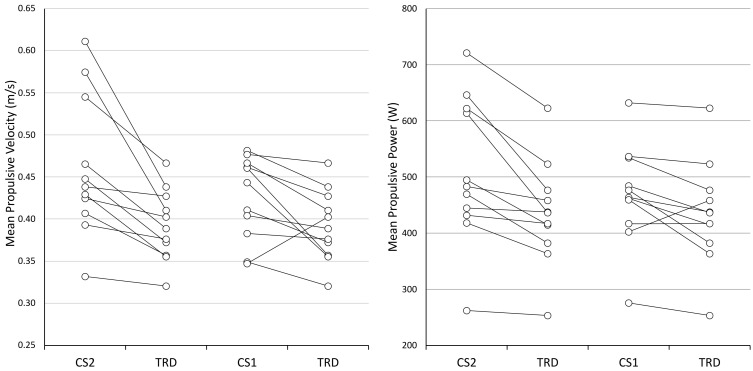
Individual measures for Mean Propulsive Velocity (**left**) and Mean Propulsive Power (**right**) for Protocols CS2 (Cluster 2), CS1 (Cluster 1), and TRD (Traditional). Data are expressed as mean value for protocol.

**Figure 4 ijerph-20-02810-f004:**
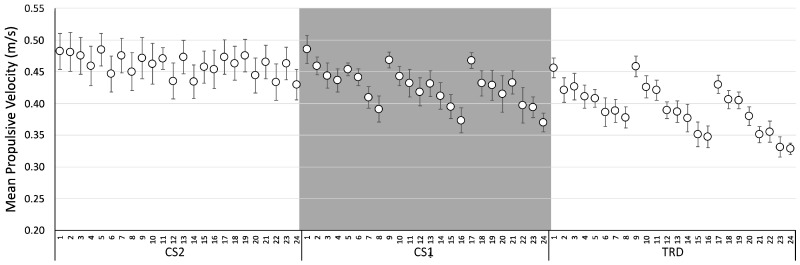
Mean Propulsive Velocity for Protocols CS2 (Cluster 2), CS1 (Cluster 1), and TRD (Traditional). Data are expressed as mean values and standard error.

**Figure 5 ijerph-20-02810-f005:**
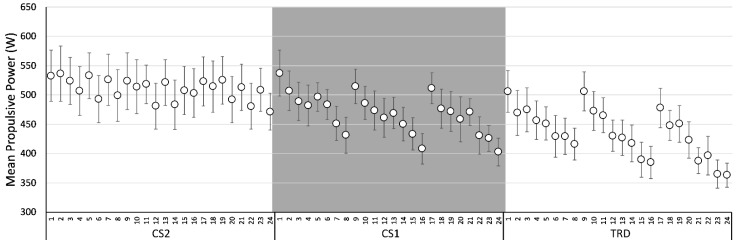
Mean Propulsive Power for Protocols CS2 (Cluster 2), CS1 (Cluster 1), and TRD (Traditional). Data are expressed as mean values and standard error.

**Figure 6 ijerph-20-02810-f006:**
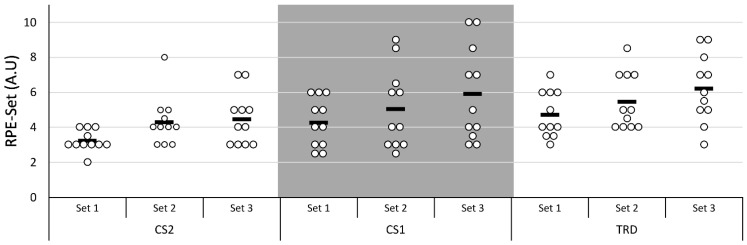
Rating of Perceived Exertion scores in arbitrary units (A.U.) for participants as individual measures (white dots) and overall sample mean (black rectangles) in the sets of each protocol.

**Table 1 ijerph-20-02810-t001:** Data (mean ± SD) comparison between protocols for all mechanical variables and their respective effect sizes.

	Protocols (Mean ± SD)			Effect Size (95% CI)		
Variables	CS2	CS1	TRD	CS2-CS1	CS2-TRD	CS1-TRD
MPV (m/s)	0.46 ± 0.08 ^TRD^	0.43 ± 0.05	0.39 ± 0.05	0.48 (−0.42–1.39)	0.99 (0.04–1.94)	0.66 (−0.26–1.57)
MPP (W)	509.75 ± 127.34 ^TRD^	467.41 ± 90.41	434.98 ± 93.82	0.37 (−0.53–1.27)	0.64 (−0.27–1.56)	0.34 (−0.56–1.24)
MPF (N)	1096.03 ± 145.26	1095.98 ± 149.54	1099.46 ± 150.11	0.00 (−0.89–0.89)	−0.02 (−0.91–0.87)	−0.02 (−0.91–0.87)
MPV Decline (%)	−5.61 ± 14.84 ^TRD, CS1^	−21.44 ± 12.13	−21.10 ± 11.88	1.12 (0.16–2.09)	1.11 (0.15–2.07)	−0.03 (−0.92–0.86)
MPP Decline (%)	−5.63 ± 14.91 ^TRD, CS1^	−21.50 ± 12.20	−20.98 ± 11.85	1.12 (0.16–2.08)	1.10 (0.14–2.06)	−0.04 (−0.93–0.85)
MPF Decline (%)	0.10 ± 0.50	−0.01 ± 1.50	0.45 ± 0.64	0.10 (−0.79–0.99)	−0.58 (−1.49–0.33)	−0.38 (−1.28–0.52)
MPV Maintenance (%)	97.74 ± 9.80 ^TRD, CS1^	90.08 ± 8.34	87.93 ± 8.54	0.81 (−0.12–1.74)	1.03 (0.08–1.98)	0.24 (−0.65–1.14)
MPP Maintenance (%)	97.76 ± 9.82 ^TRD, CS1^	89.98 ± 8.53	87.96 ± 8.55	0.81 (−0.12–1.74)	1.02 (0.07–1.98)	0.23 (−0.67–1.12)
MPF Maintenance (%)	100.12 ± 0.32	99.85 ± 1.31	100.13 ± 0.37	0.27 (−0.62–1.17)	−0.04 (−0.93–0.85)	−0.29 (−1.18–0.61)

CS2: Cluster set 2; CS1: Cluster set 1; TRD: Traditional set; MPV: Mean Propulsive Velocity; MPP: Mean Propulsive Power; MPF: Mean Propulsive Force. ^TRD^ Significantly different from TRD; ^CS1^ significantly different from CS1. *p* < 0.05.

**Table 2 ijerph-20-02810-t002:** Data (mean ± SD) comparison between protocols for all jump performance variables and their respective effect sizes.

		Protocols (Mean ± SD)			Effect Size [g, (95% CI)]		
Variables		CS2	CS1	TRD	CS2-CS1	CS2-TRD	CS1-TRD
CMJ Height (cm)	Pre	32.55 ± 6.31	32.87 ± 6.75	33.12 ± 5.99	−0.05	−0.09	−0.04
					(−0.94–0.84)	(−0.98–0.80)	(−0.93–0.85)
	0′ *	31.86 ± 6.14	31.62 ± 6.07	30.84 ± 6.09	0.04	0.16	0.12
					(−0.85–0.93)	(−0.73–1.05)	(−0.77–1.01)
	5′ **	31.17 ± 6.37	31.05 ± 6.43	31.41 ± 6.69	0.02	−0.04	−0.05
					(−0.87–0.91)	(−0.93–0.85)	(−0.94 −0.84)
	10′ **	30.87 ± 6.22	30.73 ± 6.71	30.67 ± 6.67	0.02	0.03	0.01
					(−0.87–0.91)	(−0.86–0.92)	(−0.88–0.90)
∆CMJ (%)	Pre−0′	−2.08 ± 7.11	−3.52 ± 3.47	−7.13 ± 4.70	0.25	0.81	0.84
					(−0.65–1.14)	(−0.12–1.73)	(−0.09–1.77)
	Pre−5′	−4.45 ± 5.87	−5.46 ± 4.39	−5.77 ± 4.51	0.19	0.24	0.07
					(−0.70–1.08)	(−0.65–1.13)	(−0.82–0.96)
	Pre−10′	−5.30 ± 4.40	−6.69 ± 4.72	−8.02 ± 5.56	0.29	0.52	0.25
					(−0.60–1.19)	(−0.38–1.43)	(−0.65–1.14)

CS2: Cluster set configuration 2; CS1: Cluster set configuration 1; TRD: Traditional set configuration; CMJ: countermovement jump; 0′: at the end of last set; 5′: five minutes after last set; 10′: ten minutes after last set; ∆CMJ: difference between moments; Pre-0′: difference between measures before and right after last set; Pre-5′: difference between measures before and five minutes after last set; Pre-10′: difference between measures before and ten minutes after the last set. Significantly different from Pre: * *p* < 0.05; ** *p* < 0.001.

**Table 3 ijerph-20-02810-t003:** Data (mean ± SD) comparison between protocols for RPE variables and their respective effect sizes.

	Protocols (Mean ± SD)			Effect Size [g, (95% CI)]		
Variables	CS2 ^TRD^	CS1	TRD	CS2-CS1	CS2-TRD	CS1-TRD
RPE8 (1st Set)	3.23 ± 0.61	4.27 ± 1.40	4.73 ± 1.33	−0.93 (−1.87–0.01)	−1.39 (−2.40–0.39)	−0.32 (−1.22–0.57)
RPE16 (2nd Set) *	4.32 ± 1.42	5.05 ± 2.30	5.46 ± 1.62	−0.37 (−1.27–0.53)	−0.72 (−1.64–0.20)	−0.2 (−1.09–0.69)
RPE24 (3rd Set) *^,^**	4.46 ± 1.51	5.91 ± 2.71	6.23 ± 1.97	−0.64 (−1.55–0.28)	−0.97 (−1.92–0.02)	−0.13 (−1.02–0.76)
S-RPE	4.32 ± 1.59	5.14 ± 1.85	5.68 ± 1.75	−0.46 (−1.36–0.44)	−0.78 (−1.71–0.14)	−0.29 (−1.18–0.61)

CS2: Cluster set configuration 2; CS1: Cluster set configuration 1; TRD: Traditional set configuration; RPE8: Rating of Perceived Exertion after 8th repetition; RPE16: Rating of Perceived Exertion after 16th repetition; RPE24: Rating of Perceived Exertion after 24th repetition; S-RPE: Session Rating of Perceived Exertion. ^TRD^ Significantly different from TRD, *p* < 0.01; * significantly different from 1st Set (RPE8), *p* < 0.01; ** significantly different from 2nd Set (RPE16), *p* < 0.01.

## Data Availability

The data of the present study are available on request due to privacy and ethical restrictions.

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
