# Peer review of "Influence of Cluster Sets on Mechanical and Perceptual Variables in Adolescent Athletes"

_ijerph, 2023, doi:10.3390/ijerph20042810_

Round 1
Reviewer 1 Report
Title and abstract are adequate. So as to boost the visibility of this paper in the different databases this journal is indexed in, subsitute those keywords that are already included in title. Samewise, in order to foster the current need of this study, try to update the references when possible. Provide more information about the empirical studies referred in introduction to make easier to value to what extent these studies can be considered a suitable framework for this research. In lines 442-447, review the space between lines. Discussion section is, in my humble view, perfectly developed.
Author Response
Dear Reviewer, we provide our answers in the document attached.
Please see the attachment.
Best Regards

Reviewer 2 Report
Manuscript Evaluation
Influence of Cluster Sets on Mechanical and Perceptual Variables in Adolescent Athletes
1) First, I would like to congratulate the authors considering the relevant research scope. Resistance training in adolescents is a scientific content that should be evidenced and divulged, mainly in Brazilian physical education context. In this country there is a common false consensus that mystifies and distorts physical training for children and adolescents, which harms the health and sports performance for this population. Any research that deals with this type of question is important to justify and demonstrate that physical training is suitable for this age group. Although this is not a taboo abroad, in Brazil there is a great lack in this area.
2) Even that study is from the biological and biomechanical area, it would be interesting, that the authors to address a little part of their introductory text in this sense of social importance, to clear about these problems, reenforcing the scientific contribution and paper relevance to Brazilian physical education community.
3) I think that the relevance of paper is more that the mentioned in lines 85 – 90. Please to consider the previous comment (2).
4) The general writing structure is adequate to a research manuscript. Even I don’t a native English speaker, in some reading moments I have difficult to understand. I suggest a review of language, to write with short and direct sentences. A reduction of abbreviation terms is recommendable. It is providing a friendly reading moment and better acceptance of text for physical education students and general public. A scientific paper should not knowledge only for researchers, doctors and specialists. The scientific texts should be for all people understand.
5) The figures and tables should present a smaller number of abbreviations due to technical aspect; it is improvement can provide an easily understanding for reader. Figures and tables cannot complicate the understand, in contrary should be simplify to be self-explicative.
6) In study population and sample description. In G.Power is crucial to indicate the type of test applied in present study and sample size dimension. The type of effect size description between groups is mandatory, too. Already that authors presented only 0.5 (What is it? Cohen, beta, alpha, r squared, r, t test effect size???). With these information’s, the readers are able to verify if the sample size is adequate to test. It is the unique motive to present the sample size calculi.
7) Previously, I advert according my statical experience work, the present study do not have strength to an inferential analysis, even that to consider a medium/high effect size . The use of test power as a recourse for apply a inferential statistical procedure is an common error due to hardly of “biological “sciences. We work with physical education, it is a complex knowledge, our discipline is not a basic science likely biology or basical physiology. Thus, to consider only a p-value to a N=12 in an inferential test as -anova or t test- it is not adequate. I recommend that authors do make a reading about “individual measure effects” as delta, and “individual responsiveness to physical training” is a important way to authors improve the methodology of present manuscript. For 12 participants none type statistical test for groups or a cluster of people analysis is sufficiently assertive. The p-value also is not a probability measure adequate, I recommend a reading about internal and external validity of criteria and research methods.
8) Considering also my previous comment, I see that the authors presenting the individual test effect size by formule: ΔCMJ = [(post - pre) / pre] x 100. I suggest that results can be centered in this analysis. It is most important than p-value or means between groups.
9) Lines 395-406: I am uncertain whether another reader of the study would be able to understand such a large number of numbers and information. I apologize for my bluntness; however, I have to say that it is extremely exhausting looking for results in tables, in this tangle of numbers and letters.
10) I suggest that the authors focus the study result on 1 type of effect and the main results.
11) In scientific research we do not need to disclose all applied statistics. Only what is pertinent and relevant to demonstrate the result.
12) I am sorry, however in lines 491-494 my reading ability has run out. It is impossible to a people enjoy and understand a paper that present a text as follow:
“Our results showed that MPV and MPP were different only for CS2 in comparison to TRD but not for CS2 x CS1 and CS1 x TRD. As well, MVD, MPD, MVM, and MPM pre-sented smaller decreases and better maintenance, especially when more intra-set rest intervals were applied on the sets, showing that CS2 was superior to CS1 and TRD.”
13) With extreme respect to you know how about physical training. I ask you, about upper cited sentence:
What is it? MPV, MPP, CS2, TRD, CS1,MVM, MPM?
Are you really credible that people understand or will find in previous text information the significate of abbreviations? I recommend a deeper reflexing about the text quality of presentation and readability.
14) Considering the general research methodology and results, my suggestion is changing the study design to a “multiple individual cases study with two nonequivalent groups”, focused in describe the effect size in each one of participants and details that present clinical crossover design do not permit. Mainly, in view of the first principle of physical training theory: biological individuality.
Author Response

(The authors gave the same response as above.)

Reviewer 3 Report
Thanks for giving me this opportunity to review this paper. The present study aimed to compare the effect of cluster sets (CS) on the performance of mechanical and perceptual variables in young athletes. The results suggest that using CS with a greater number of intra-set rests is more efficient even with the total rest interval equalized, presenting lower decreases in mechanical performance and lower perceptual effort responses. Generally, the research topic is interesting and the manuscript is well prepared. I have several concerns that may be considered by the authors to further improve their paper. See the below comments for the detail.
1. The abstract, results, better to highlight some values of primary outcomes rather than only mentioning the p-value.
2. The rationale of the present study needs to be further highlighted. Currently, only one sentence is used to mention the rationale (line 85-86) that is not enough.
3. Line 92-96, should be removed from the introduction as it belongs to the methods section.
4. Figure 1 can be further improved as all the contents are the same under 2nd day, 3rd day, and 4th day. Therefore, it can be further simplified.
5. “Materials and methods”, may mention participants first, followed by the study design and context.
6. One important limitation is that the exercise training and/or nutrition status between the two trials were not controlled, or at least were not reported. These factors may affect the final results, and should be mentioned.
Author Response

(The authors gave the same response as above.)

Round 2
Reviewer 2 Report
The authors provide sufficient corrections and explanations.
Thank you very much by this invite.